# GETMusic: Generating Music Tracks with a Unified Representation and Diffusion Framework

## Abstract

Symbolic music generation aims to create musical notes, which can help users compose music, such as generating target instrument tracks based on provided source tracks. In practical scenarios where there's a predefined ensemble of tracks and various composition needs, an efficient and effective generative model that can generate any target tracks based on the other tracks becomes crucial. However, previous efforts have fallen short in addressing this necessity due to limitations in their music representations and models. In this paper, we introduce a framework known as GETMusic, with "GET" standing for "GEnerate music Tracks." This framework encompasses a novel music representation "GETScore" and a diffusion model "GETDiff." GETScore represents musical notes as tokens and organizes tokens in a 2D structure, with tracks stacked vertically and progressing horizontally over time. At a training step, each track of a music piece is randomly selected as either the target or source. The training involves two processes: In the forward process, target tracks are corrupted by masking their tokens, while source tracks remain as the ground truth; in the denoising process, GETDiff is trained to predict the masked target tokens conditioning on the source tracks. Our proposed representation, coupled with the non-autoregressive generative model, empowers GETMusic to generate music with any arbitrary source-target track combinations. Our experiments demonstrate that the versatile GETMusic outperforms prior works proposed for certain specific composition tasks. Our music demos are available at https://getmusicdemo.github.io/. Our code is in the supplementary materials and will be open.

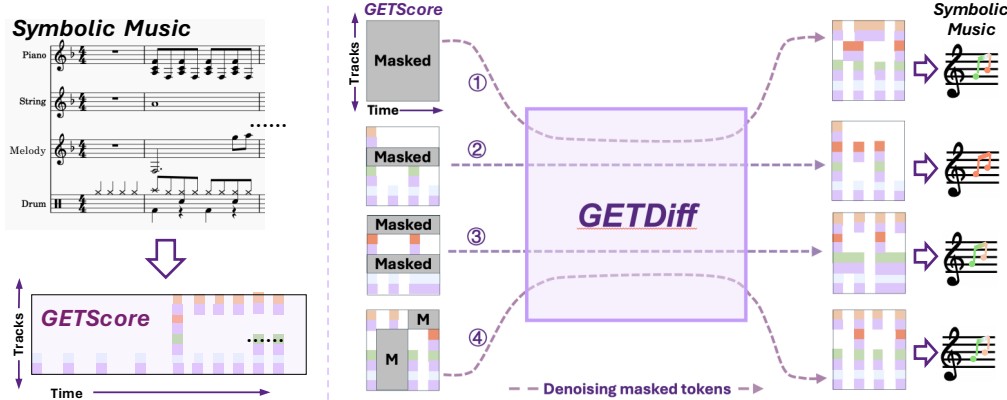

Figure 1: The overview of GETMusic, involving a novel music representation "GETScore" and a discrete diffusion model "GETDiff." Given a predefined ensemble of instrument tracks, GETDiff takes GETScores as inputs and can generate any desired target tracks conditioning on any source tracks (①, ②, and ③). This flexibility extends beyond track-wise generation, as it can perform zero-shot generation for any masked parts (④).

# 1 INTRODUCTION

Symbolic music generation aims to create musical notes, which can help users in music composition. Due to the practical need for flexible and diverse music composition, the need for an efficient and unified approach capable of generating arbitrary tracks based on the others is high[1]. However, current research falls short of meeting this demand due to inherent limitations imposed by their representations and models. Consequently, these approaches are confined to specific source-target combinations, such as generating piano accompaniments based on melodies.

Current research can be categorized into two primary approaches based on music representation: sequence-based and image-based. On one hand, sequence-based works (Huang & Yang, 2020; Zeng et al., 2021; Christopher, 2011) represent music as a sequence of discrete tokens, where a musical note requires multiple tokens to describe attributes such as onset, pitch, duration, and instrument. These tokens are arranged chronologically, resulting in the interleaving of notes from different tracks, and are usually predicted by autoregressive models sequentially. The interleaving of tracks poses a challenge of precise target generation because the autoregressive model implicitly determines when to output a target-track token and avoids generating tokens from other tracks. It also complicates the specification of source and target tracks. Therefore, the existing methods (Dong et al., 2023; Ren et al., 2020; Yu et al., 2022) typically focus on either one specific source-target track combination or the continuation of tracks.

On the other hand, image-based research represents music as 2D images, with pianorolls[2] being a popular choice. Pianorolls represent musical notes as horizontal lines, with the vertical position denoting pitch and the length signifying duration. A pianoroll explicitly separates tracks but it has to incorporate the entire pitch range of instruments, resulting in large and sparse images. Due to the challenges of generating sparse and high-resolution images, most research has focused on conditional composition involving only a single source or target track (Dong et al., 2017; Yang et al., 2017; Shuyu & Sung, 2023) or unconditional generation (Mittal et al., 2021).

To support the generation across flexible and diverse source-target track combinations, we propose a unified representation and diffusion framework called GETMusic ("GET" stands for **GE**nerate music **T**racks), which comprises a representation named GETScore, and a discrete diffusion model (Austin et al., 2021) named GETDiff. GETScore represents the music as a 2D structure, where tracks are stacked vertically and progress horizontally over time. Within each track, we efficiently represent musical notes with the same onset by a single pitch token and a single duration token, and position them based on the onset time. At a training step, each track in a training sample is randomly selected as either the target or the source. The training consists of two processes: In the forward process, the target tracks are corrupted by masking tokens, while the source tracks are preserved as ground truth; in the denoising process, GETDiff learns to predict the masked target tokens based on the provided source. Our co-designed representation and diffusion model in GETMusic offer several advantages compared to prior works:

• With separate and temporally aligned tracks in GETScore, coupled with a non-autoregressive generative model, GETMusic adeptly compose music across various source-target combinations.

• GETScore is a compact multi-track music representation while effectively preserving interdependencies among simultaneous notes both within and across tracks, fostering harmonious music generation.

• Beyond track-wise generation, the mask and denoising mechanism of GETDiff enable the zero-shot generation (i.e., denoising masked tokens at any arbitrary locations in GETScore), further enhancing the versatility and creativity.

In this paper, our experiments consider six instruments: *bass, drum, guitar, piano, string*, and *melody*, resulting in 665 source-target combinations (Details in appendix A). We demonstrate that our proposed versatile GETMusic surpasses approaches proposed for specific tasks such as conditional accompaniment or melody generation, as well as generation from scratch.

---

[1]A music typically consists of multiple instrument tracks. In this paper, given a predefined track ensemble, we refer to the tracks to be generated as "target tracks" and those acting as conditions as "source tracks." We refer to such an orchestration of tracks as a "source-target combination."

[2]https://en.wikipedia.org/wiki/Piano_roll

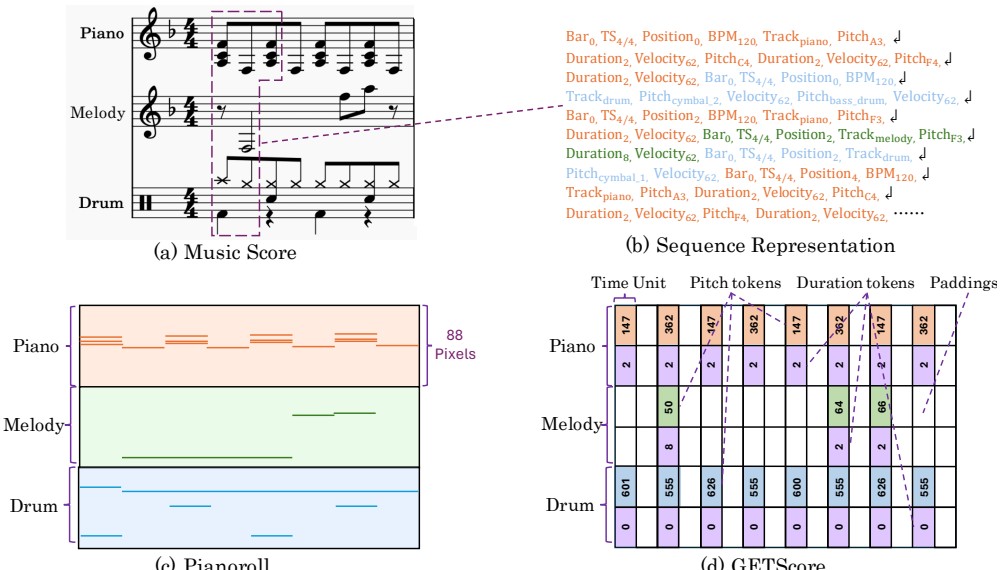

Figure 2: Different representations for the same piece of music. Figure (a) is the music score. Figure (b) illustrates the sequence-based representation in REMI (Huang & Yang, 2020) style, and due to the length of the sequence, we only show the portion enclosed by the dashed box in Figure (a). Figure (c) shows a sparse pianoroll that represents notes by lines. In Figure (d), GETScore separates and aligns tracks, forming the basis for unifying generation across various source-target combinations. It also preserves the interdependencies among simultaneous notes, thereby fostering harmony in music generation. Numbers in (d) denote token indices which are for demonstration only.

## 2 BACKGROUND

### 2.1 SYMBOLIC MUSIC GENERATION

Symbolic music generation aims to generate musical notes, whether from scratch (Mittal et al., 2021; Yu et al., 2022) or based on given conditions such as chords, tracks (Shuyu & Sung, 2023; Huang & Yang, 2020; Dong et al., 2017), lyrics (Lv et al., 2022; Ju et al., 2021; Sheng et al., 2020), or other musical properties (Zhang et al., 2022), which can assist users in composing music. In practical music composition, a common user need is to create instrumental tracks from scratch or conditioning on existing ones. Given a predefined ensemble of tracks and considering flexible composition needs in practice, a generative model capable of handling arbitrary source-target combination is crucial. However, neither of the existing approaches can integrate generation across multiple source-target combinations, primarily due to inherent limitations in their representations and models.

Current approaches can be broadly categorized into two main categories with respect to adopted representation: sequence-based and image-based. In sequence-based methods (Huang & Yang, 2020; Hsiao et al., 2021; Zeng et al., 2021; Ren et al., 2020), music is represented as a sequence of discrete tokens. A token corresponds to a specific attribute of a musical note, such as onset (the beginning time of a note), pitch (note frequency), duration, and instrument, and tokens are usually arranged chronologically. Consequently, notes that represent different tracks usually interleave, as shown in Figure 2(b) where the tracks are differentiated by colors. Typically, an autoregressive model is applied to processes the sequence, predicting tokens one by one. The interwove tracks and the autoregressive generation force the model to implicitly determine when to output tokens of desired target tracks and avoid incorporating tokens belonging to other tracks, which poses a challenge to the precise generation of the desired tracks; the sequential representation and modeling do not explicitly preserve the interdependencies among simultaneous notes, which impact the harmony of the generated music; furthermore, the model is required to be highly capable of learning long-term dependencies (Bengio et al., 1994) given the lengthy sequences. Some unconventional methods (Ens & Pasquier, 2020) organize tokens according to the track order in order to eliminate track interleaving. However, it comes with a trade-off, as it results in weaker dependencies both in the long term and across tracks.

Image-based methods mainly employ pianoroll representations which depict notes as horizontal lines in 2D images, with the vertical position denoting pitch and the length signifying duration. However, pianorolls need to include the entire pitch range of the instrument, resulting in images that are both large and sparse. For instance, Figure 2(c) illustrates a pianoroll representation of a three-track music piece, which spans a width of hundreds of pixels, yet only the bold lines within it carry musical information. Most works focus on conditional composition involving only a single source/target track (Dong et al., 2017; Yang et al., 2017; Shuyu & Sung, 2023) or unconditional generation (Mittal et al., 2021) because generating a sparse and high-resolution image is challenging.

Our proposed GETMusic addresses above limitations with a co-designed representation and a discrete diffusion model which together provide an effective solution to versatile track generation.

## 2.2 DIFFUSION MODELS

Diffusion models, initially proposed by (Sohl-Dickstein et al., 2015) and further improved by subsequent research (Ho et al., 2020; Song et al., 2021; Ho & Salimans, 2021; Dhariwal & Nichol, 2021), have demonstrated impressive capabilities in modeling complex distributions. These models consist of two key processes: a forward (diffusion) process and a reverse (denoising) process. The forward process $q(x_{1:T}|x_0) = \prod_{t=1}^{T} q(x_t|x_{t-1})$ introduces noise to the original data $x_0$ iteratively for $T$ steps, corrupting it towards a prior distribution $p(x_T)$ that is independent of $x_0$. The goal of diffusion models is to learn a reverse process $p_\theta(x_{t-1}|x_t)$ that gradually denoises $x_T$ to the data distribution. The model is trained by optimizing the variational lower bound (VLB) (Ho et al., 2020):

$$L_{\text{vlb}} = \mathbb{E}_q[-\log p_\theta(x_0|x_1)] + \sum_{t=2}^{T} D_{KL}\left[q(x_{t-1}|x_t, x_0)||p_\theta(x_{t-1}|x_t))\right] + D_{KL}[q(x_T|x_0)||p(x_T)]]. \quad (1)$$

Diffusion models can be categorized into continuous and discrete versions. As our proposed GETScore represents music as a 2D arrangement of discrete tokens, we employ the discrete diffusion framework in our method. Discrete diffusion models in (Sohl-Dickstein et al., 2015) were developed for binary sequence learning. Hoogeboom et al. (2021) extended these models to handle categorical random variables, while Austin et al. (2021) introduced a more structured categorical forward process: the forward process is a Markov chain defined by transition matrices, which transitions a token at time $t-1$ to another at time $t$ by probability. For our diffusion model GETDiff, we adopt their forward process as the basis. We also adopt a crucial technique known as $x_0$-parameterization (Austin et al., 2021), where instead of directly predicting $x_{t-1}$ at time step $t$, the model learns to fit the noiseless original data $x_0$ and corrupts the predicted $\tilde{x}_0$ to obtain $x_{t-1}$. Consequently, an auxiliary term scaled by a hyper-parameter $\lambda$ is added to the VLB:

$$L_\lambda = L_{\text{vlb}} + \lambda \mathbb{E}_q\left[\sum_{t=2}^{T} -\log p_\theta(x_0|x_t)\right] \quad (2)$$

## 3 GETMUSIC

In this section, we introduce two key components in GETMusic: the representation GETScore and the diffusion model GETDiff. We first provide an overview of each component, and then highlight their advantages in supporting the flexible and diverse generation of any tracks.

## 3.1 GETSCORE

Our goal is to design an efficient and effective representation for modeling multi-track music, which allows for flexible specification of source and target tracks and thereby laying the foundation of the diverse track generation tasks. Our novel representation GETScore involves two core ideas: (1) the 2D track arrangement and (2) the musical note tokenization.

**Track Arrangement** We derive inspiration from music scores to arrange tracks vertically, with each track progressing horizontally over time. The horizontal axis is divided into fine-grained temporal units, with each unit equivalent to the duration of a 16th note. This level of temporal detail is sufficient to the majority of our training data. This arrangement of tracks brings several benefits:

- It prevents content of different tracks from interleaving, which simplifies the specification of source and target tracks, and facilitates the precise generation of desired tracks.

- Because tracks are temporally aligned like music scores, their interdependencies are well preserved.

**Note Tokenization**    To represent musical notes, we focus on two attributes: pitch and duration, which are directly associated with composition. Some dynamic factors like velocity and tempo variation fall outside the scope of our study. We use two distinct tokens to denote a note's pitch and duration, respectively. These paired pitch-duration tokens are placed in accordance with the onset time and track within GETScore. Some positions within GETScore may remain unoccupied by any tokens; in such instances, we employ padding tokens to fill them, as illustrated by the blank blocks in Figure 2(d). Each track has its own pitch token vocabulary but shares a common duration vocabulary, considering pitch characteristics are instrument-dependent, whereas duration is a universal feature across all tracks. To broaden the applicability of GETScore, we need to address two more problems:

(1) How to use single pitch and duration tokens to represent a group of notes played simultaneously *within a track*? We propose merging pitch tokens of a group of simultaneous notes into a single compound pitch token. Furthermore, we identify the most frequently occurring duration token within the group as the final duration token. This simplification of duration representation is supported by our observation from the entire training data, where notes in more than 97% groups share the same duration. In only 0.5% groups, the maximum duration difference among notes exceeds a temporal unit. These findings suggest that this simplification has minimal impact on the expressive quality of GETScore. Figure 2(d) illustrates the compound token: in the piano track, we merge the first three notes "A", "C", and "F" into a single token indexed as "147."

(2) How to represent percussive instruments, such as drums, which do not involve the concepts of "pitch" and "duration?" We treat individual drum actions (e.g., kick, snare, hats, toms, and cymbals) as pitch tokens and align them with a special duration token. The drum track in Figure 2(d) illustrates our approach.

In conclusion, besides the benefits from track arrangement, GETScore also gains advantages through this note tokenization:

- Each track requires only two rows to accommodate the pitch and duration tokens, significantly enhancing the efficiency of GETScore.

- The compound token preserves the interdependecies within a track. When it is generated, harmony is inherently guaranteed because the corresponding note group is derived from real-world data.

## 3.2   GETDIFF

In this section, we first introduce the forward and the denoising process of GETDiff during training, respectively. Next, we introduce the inference procedure and outline GETDiff's benefits in addressing the diverse needs for track generation.

**The Forward Process**    Since GETMusic operates on GETScore, which consists of discrete tokens, we employ a discrete diffusion model. We introduce a special token [MASK] into the vocabulary as the absorbing state of the forward process. At time $t - 1$, a normal token remains in its current state with a probability of $\alpha_t$ and transitions to [MASK] (i.e., corrupts to noise) with a probability of $\gamma_t = 1 - \alpha_t$. As GETScore includes a fixed number of tracks that GETMusic supports, and the composition does not always involve all tracks, we fill the uninvolved tracks with another special token [EMPTY]. [EMPTY] never transitions to other tokens, nor can it be transitioned to from any other tokens. This design prevents any interference from uninvolved tracks in certain compositions. Formally, a transition matrix $[Q_t]_{mn} = q(x_t = m | x_{t-1} = n) \in \mathbb{R}^{K \times K}$ defines the transition probability from the $n$-th token at time $t - 1$ to the $m$-th token at time $t$:

$$
Q_t = \begin{bmatrix}
\alpha_t & 0 & 0 & \dots & 0 & 0 \\
0 & \alpha_t & 0 & \dots & 0 & 0 \\
0 & 0 & \alpha_t & \dots & 0 & 0 \\
\vdots & \vdots & \vdots & \ddots & \vdots & \vdots \\
0 & 0 & 0 & \dots & 1 & 0 \\
\gamma_t & \gamma_t & \gamma_t & \dots & 0 & 1
\end{bmatrix}, \tag{3}
$$

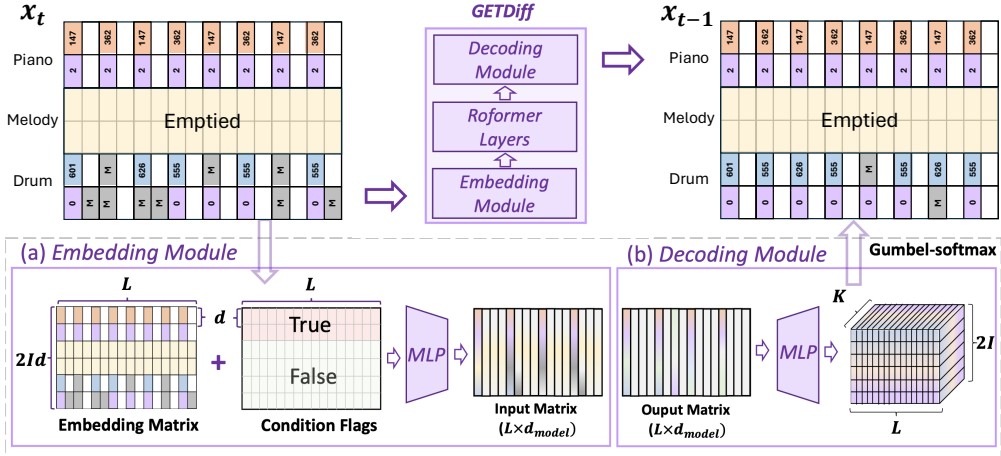

Figure 3: An overview of training the GETDiff using a 3-track GETScore. Note that GETScore is capable of accommodating any number of tracks, with this example serving as a toy example. During this training step, GETMusic randomly selects the piano track as the source and the drum track as the target, while ignoring the melody track. Thus, $x_t$ consists of the ground truth piano track, an emptied melody track, and a corrupted drum track. GETDiff generates all tokens simultaneously in a non-autoregressive manner which may modify tokens in its output. Therefore, when $x_{t-1}$ is obtained, the sources are recovered with the ground truth while ignored tracks are emptied again.

where $K$ is the total vocabulary size, including two special tokens. The last two columns of the matrix correspond to $q(x_t|x_{t-1} = \texttt{[EMPTY]})$ and $q(x_t|x_{t-1} = \texttt{[MASK]})$, respectively. Denoting $v(x)$ as a one-hot column vector indicating the category of $x$ and considering the Markovian nature of the forward process, we can express the marginal at time $t$, and the posterior at time $t-1$ as follows:

$$q(x_t|x_0) = v^\top(x_t)\overline{Q}_t v(x_0), \quad \text{with} \quad \overline{Q}_t = Q_t \dots Q_1. \tag{4}$$

$$q(x_{t-1}|x_t, x_0) = \frac{q(x_t|x_{t-1}, x_0)q(x_{t-1}|x_0)}{q(x_t|x_0)} = \frac{\left(v^\top(x_t)Q_t v(x_{t-1})\right)\left(v^\top(x_{t-1})\overline{Q}_{t-1}v(x_0)\right)}{v^\top(x_t)\overline{Q}_t v(x_0)}. \tag{5}$$

With the tractable posterior, we can optimize GETDiff with Eq.2.

**The Denoising Process** Figure 3 provides an overview of GETMusic denoising a three-track training sample of a length of $L$ time units. GETDiff has three main components: an embedding module, Roformer (Su et al., 2021) layers, and a decoding module. Roformer is a transformer (Vaswani et al., 2017) variant that incorporates relative position information into the attention matrix, which enhances the model's ability to length extrapolation during inference.

During training, GETMusic needs to cover the various source-target combinations for a music piece with $I$ tracks, represented as a GETScore with $2I$ rows. To achieve this, $m$ tracks (resulting in $2m$ rows in GETScore) are randomly chosen as the source, while $n$ tracks (resulting in $2n$ rows in GETScore) are selected as the target, $m \geq 0$, $n > 0$, and $m + n \leq I$.

At a randomly sampled time $t$, to obtain $x_t$ from the original GETScore $x_0$, tokens in target tracks are transitioned according to $\overline{Q}_t$, tokens in the source tracks remain as the ground truth, and uninvolved tracks are emptied. GETDiff denoises $x_t$ in four steps, as shown in Figure 3: (1) All tokens in GETScore are embedded into $d$-dimensional embeddings, forming an embedding matrix of size $2Id \times L$. (2) Learnable condition flags are added in the matrix to guide GETDiff which tokens can be conditioned on, thereby enhancing inference performance. The effectiveness of condition flags is analyzed in § 4.3. (3) The embedding matrix is resized to GETDiff's input dimension $d_{model}$ using an MLP, and then fed into the Roformer model. (4) The output matrix passes through a classification head to obtain the token distribution over the vocabulary of size $K$ and we obtain the final tokens using the gumbel-softmax technique.

**Inference** During inference, users can specify any target and source tracks, and GETMusic constructs the corresponding GETScore representation, denoted as $x_T$, which contains the ground truth of source tracks, masked target tracks, and emptied tracks (if any). GETMusic then denoises $x_T$

step by step to obtain $x_0$. As GETMusic generates all tokens simultaneously in a non-autoregressive manner, potentially modifying source tokens in the output, we need to ensure the consistent guidance from source tracks: when $x_{t-1}$ is acquired, tokens in source tracks are recovered to their ground truth values, while tokens in uninvolved tracks are once again emptied.

Considering the combined benefits of the representation and the diffusion model, GETMusic offers two major advantages in addressing the diverse composition needs:

• Through a unified diffusion model, GETMusic has the capability to compose music across a range of source-target combinations without requiring re-training.

• Beyond the track-wise generation, the mask and denoising mechanism of GETDiff enables the zero-shot generation of any arbitrary masked locations in GETScore, which further enhances versatility and creativity. An illustration of this can be found in case ④ in Figure 1.

## 4 EXPERIMENTS

### 4.1 EXPERIMENT SETTINGS

**Data and Preprocess** We crawled 1,569,469 MIDI files from Musescore[3]. We followed Ren et al. (2020) to pre-process the data, resulting in music including $I = 6$ instrumental tracks: *bass, drum, guitar, piano, string, melody* and an extra chord progression track. After strict cleanse and filter, we construct 137,812 GETScores (about 2,800 hours) with the maximum $L$ as 512, out of which we sampled 1,000 for validation, 100 for testing, and the remaining for training. We train all baselines on the crawled data. The vocabulary size $K$ is 11,883. More details on data and the pre-processing are in appendix B.

**Training Details** We set diffusion timesteps $T = 100$ and the auxiliary loss scale $\lambda = 0.001$. For the transition matrix $Q_t$, we linearly increase $\overline{\gamma}_t$ (cumulative $\gamma_t$) from 0 to 1 and decrease $\overline{\alpha}_t$ from 1 to 0. GETDiff has 12 Roformer layers with $d = 96$ and $d_{model} = 768$, where there are about 86M trainable parameters. During training, we use AdamW optimizer with a learning rate of $1e - 4$, $\beta_1 = 0.9$, $\beta_2 = 0.999$. The learning rate warmups first 1000 steps and then linearly decays. The training is conducted on $8 \times 32G$ Nvidia V100 GPUs and the batch size on each GPU is 3. We train the model for 50 epochs and validate it every 1000 steps, which takes about 70 hours in total. We select model parameters based on the validation loss.

**Tasks and Baselines** We consider three symbolic music generation tasks: (1) *accompaniment generation based on the melody*, (2) *melody generation based on the accompaniments*, and (3) *generating tracks from scratch*. For the first two tasks, we compare GETMusic with PopMAG (Ren et al., 2020). PopMAG is an autoregressive transformer encoder-decoder model that processes a sequence representation MuMIDI. Following Ren et al. (2020), an extra chord progression provides more composition guidance and we treat the chord progression as another track in GETScore (Details in appendix B). To be comparable, we restrict the generated music to a maximum length of 128 beats, which is the longest composition length for PopMAG. For the third task, we compare GETMusic with Museformer (Yu et al., 2022), one of the most competitive unconditional generation models. We generate all 6 tracks of 100 songs from scratch, where each song also restricted to 128 beats.

**Evaluation** We introduce objective metrics that quantitatively evaluates the generation quality. Following Ren et al. (2020), we evaluate the models from two aspects:

(1) *Chord Accuracy*: For Task 1 and 2, we measure the chord accuracy *CA* between generated target tracks and their ground truth to evaluate the melodic coherence:

$$CA = \frac{1}{N_{tracks} * N_{chords}} \sum_{i=1}^{N_{tracks}} \sum_{j=1}^{N_{chords}} \mathbb{1}(C'_{i,j} = C_{i,j}). \tag{6}$$

Here, $N_{tracks}$ and $N_{chords}$ represent the number of tracks and chords, respectively. $C'_{i,j}$ and $C_{i,j}$ denote the $j$-th chord in the $i$-th generated target track and the ground truth, respectively. Note that

---

[3]https://musescore.com/

Table 1: We compare GETMusic with PopMAG and Museformer, through three representative tasks: the accompaniment/melody generation as well as generating from scratch. In all human evaluations, the $\kappa$ values consistently exceed 0.6, indicating substantial agreement among the evaluators.

| Method | $CA(\%)\uparrow$ | $KL_{Pitch}\downarrow$ | $KL_{Dur}\downarrow$ | $KL_{IOI}\downarrow$ | $HR\uparrow$ |
|---|---|---|---|---|---|
| | | Accompaniment Generation | | | |
| PopMAG | 61.17 | 10.98 | 7.00 | 6.92 | 2.88 |
| **GETMusic** | **65.48** | **10.05** | **4.21** | **4.22** | **3.35** |
| | | Lead Melody Generation | | | |
| PopMAG | 73.70 | 10.64 | 3.97 | 4.03 | 3.14 |
| **GETMusic** | **81.88** | **9.82** | **3.67** | **3.49** | **3.52** |
| | | Generation from Scratch | | | |
| Museformer | - | 8.19 | **3.34** | 5.71 | 3.05 |
| **GETMusic** | - | **7.99** | 3.38 | **5.33** | **3.18** |

this metric is not suitable for the third task. Instead, melodic evaluation for the third task relies on both the pitch distribution and human evaluation, which are discussed later.

(2) *Feature Distribution Divergence*: For the first two tasks, we assess the distributions of some important musical features in generated and ground truth tracks: note pitch, duration (*Dur*) and Inter-Onset Interval (*IOI*) that measures the temporal interval between two consecutive notes within a bar. First, we quantize the note pitch, duration and *IOI* into 16 classes, then convert the histograms into probability density functions (PDFs) using Gaussian kernel density estimation. Finally, we compute the KL-divergence (Kullback & Leibler, 1951) $KL_{\{Pitch,Dur,IOI\}}$ between the PDFs of generated target tracks and ground truth. For the third task, we compute $KL_{\{Pitch,Dur,IOI\}}$ between the PDFs of generated target tracks and the corresponding distribution of training data.

(4) *Human Evaluation*: We recruited 10 evaluators with basic music knowledge. They were presented with songs generated by GETMusic and baselines in a blind test. Evaluators provided a *Human Rating (HR)*, on a scale from 1 (Poor) to 5 (Excellent). The HR rating reflects the overall quality of the generated songs, and the coherence between the target and source tracks (when applicable). More details on human evaluation are in appendix C.

## 4.2 GENERATION RESULTS

**Comparison with Previous Works** Table 1 presents the results of three composition tasks. In the first two tasks, GETMusic consistently outperforms PopMAG across all metrics, showcasing its ability to create music with more harmonious melodies and rhythms that align well with the provided source tracks. When we compare the first two tasks, an improvement in music quality becomes evident as we involve more source tracks. In the second task, where all five accompaniment instruments serve as source tracks, we achieve better scores in most metrics compared to the first task which relies solely on the melody as the source track. In unconditional generation, GETMusic outperforms the competitive baseline in most metrics. Subjective evaluations further confirm the effectiveness of GETMusic. Readers are welcome to visit our demo page for generated samples.

**Zero-shot Generation** Although GETMusic is trained for track-wise generation, it can zero-shot recover masked tokens at any arbitrary locations, due to its the mask and denoising mechanism. The zero-shot generation is examplified in case ④ in Figure 1. This capability enhances the versatility and creativity of GETMusic. For example, we can insert mask tokens in the middle of two different songs to connect them: GETMusic generates a harmonious bridge by iteratively denoising the masked tokens while preserving the rest of the tokens unchanged. Despite the challenges in evaluation, the 8th and 9th demos on the demo page showcase our approach's flexibility and creativity.

## 4.3 METHOD ANALYSIS

**The Complementary Nature of GETScore and GETDiff** To demonstrate this, we begin with an ablation study in which we replace GETDiff with an autoregressive model. For the task of generating

Table 2: Ablation study on generation paradigms: Autoregressive vs. Non-autoregressive.

| Method | $CA(\%) \uparrow$ | $KL_{Pitch} \downarrow$ | $KL_{Dur} \downarrow$ | $KL_{IOI} \downarrow$ | $Time \downarrow$ | $HR \uparrow$ |
|---|---|---|---|---|---|---|
| PopMAG | 61.17 | 10.98 | 7.00 | 6.92 | 23.32 | 2.88 |
| GETMusic (AR) | 46.25 | 11.91 | 7.08 | 6.49 | 17.13 | 2.37 |
| **GETMusic** | **65.48** | **10.05** | **4.21** | **4.22** | **4.80** | **3.35** |

Table 3: Ablation study on the effectiveness of condition flags.

| Method | $CA \uparrow$ | $KL_{Pitch} \downarrow$ | $KL_{Dur} \downarrow$ | $KL_{IOI} \downarrow$ | $Loss \downarrow$ |
|---|---|---|---|---|---|
| **GETMusic** (AG) | **65.48** | **10.05** | **4.21** | **4.22** | **1.39** |
| $-$ condition flags | 45.16 | 10.89 | 6.32 | 5.34 | 1.40 |
| **GETMusic** (UN) | - | **7.99** | **3.38** | **5.33** | **1.63** |
| $-$ condition flags | - | 8.43 | 3.57 | 5.61 | 1.75 |

music from scratch, we train a transformer decoder equipped with 12 prediction heads. At each decoding step, it predicts 12 tokens (6 pitch tokens and 6 duration tokens in a GETScore involving 6 tracks). The outcomes of this variant, denoted as GETMusic (AR), are detailed in Table 2, revealing suboptimal results characterized by a tendency to produce repetitive melody. Additionally, we present the average time required in seconds for composing each musical piece using an Nvidia A100 GPU, highlighting that the non-autoregressive denoising process significantly outpaces autoregressive decoding in terms of speed.

While it would be more informative to evaluate diffusion models trained with traditional sequence representations, this approach is intractable. Firstly, due to the inherently higher computational resource requirements of training a diffusion model compared to an autoregressive model, coupled with the fact that traditional sequence representations are typically an order of magnitude longer than GETScore when representing the same musical piece, the training cost becomes prohibitively high. Furthermore, diffusion models require the specification of the generation length in advance. Yet, the length of traditional sequences representing the same number of bars can vary in a wide range, leading to uncontrollable variations in the generated music's length and structure.

Based on above results and analyses, we believe that our proposed GETScore and GETDiff together provide an efficient and effective solution for versatile and diverse symbolic music generation.

**Effectiveness of Condition Flags** In GETScore, since all normal tokens carry information, any inaccuracies in the predicted normal tokens can lead to deviations in the denoising direction during inference. To address this issue, we incorporate learnable condition flags into the embedded GETScore to signify trustworthy tokens. To evaluate the effectiveness of the condition flags, we remove them from the diffusion model. The results are shown in Table 3. Given the comparable loss, removing the condition flags has minimal impact on training and convergence, but it leads to lower generation quality in accompaniment generation (AG) while slightly affecting unconditional generation (UN). This demonstrates the effectiveness of condition flags in guiding the model to generate high-quality music, particularly in conditional generation scenarios.

## 5 CONCLUSION

We propose GETMusic, a unified representation and diffusion framework to effectively and efficiently generate desired target tracks from scratch or based on user-provided source tracks, which can address users' diverse composition needs. GETMusic has two core components: a novel representation GETScore and a diffusion model GETDiff. GETScore offers several advantages in representing multi-track music, including efficiency, simple source-target specification, and explicit preservation of simultaneous note interdependencies. Leveraging the power of GETScore and the non-autoregressive nature of GETDiff, GETMusic can compose music across various source-target combinations and perform zero-shot generation at arbitrary locations. In the future, we will continue to explore the potential of GETMusic, such as incorporating lyrics as a track to enable lyric-to-melody generation.

## REPRODUCIBILITY

To ensure the reproducibility of our work, apart from providing a comprehensive description of our proposed representation (§3.1) and model (§3.2), we have included detailed implementation information in appendix B. This includes specifics on data cleansing, the incorporation of chord progression as a condition, and the construction of the vocabulary. Additionally, we have made our code available in the supplementary materials, and we are committed to open-sourcing our code, preprocessing scripts, and model checkpoints.

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

# A  THE NUMBER OF SOURCE-TARGET DIVSIONS

For a given $k$-track music input, GETMusic can select $m$ tracks as the source and generate $n$ target tracks selected from the remaining $k - m$ tracks. Considering that each track can be used as the source, target, or left empty, there are $3^k$ possible combinations. However, a specific scenario is that $m$ tracks are selected as the source and leaving the remaining $k - m$ tracks empty, resulting in $\sum_{m=0}^{k} C_k^m = 2^k$ illegal combinations. Therefore, the number of valid combinations is $3^k - 2^k$.

In our setting, we have six instrumental tracks, resulting in 665 possible combinations. Notably, the chord progression track is not considered as 7-th track in this calculation because we consistently enable chord progression as a source track to enhance the quality of conditional generation.

# B  DATA PRE-PROCESSING

**Cleanse Data**  Following the method proposed in (Ren et al., 2020), we perform a data cleansing process by four steps. Firstly, we employ MIDI Miner (Guo et al., 2019) to identify the melody track. Secondly, we condense the remaining tracks into five instrument types: *bass*, *drum*, *guitar*, *piano*, and *string*. Thirdly, we apply filtering criteria to exclude data that contains a minimal number of notes, has less than 2 tracks, exhibits multiple tempos, or lacks the melody track. Fourthly, for all the data, we utilize the Viterbi algorithm implemented by Magenta (`https://github.com/magenta/magenta`) to infer the corresponding chord progression, which serves as an additional composition guide. Lastly, we segment the data into fragments of up to 32 bars and convert these fragments into GETScore representation.

**Chord Progression**  The configuration of the chord progression track is different from regular instrumental tracks. Although certain commonly used chords may appear in specific instrumental tracks and have been represented as pitch tokens, we do not reuse these tokens to ensure that the chord progression track provides equitable guidance for each individual track.

GETMusic incorporates 12 chord roots: `C, C#, D, D#, E, F, F#, G, G#, A, A#, B` and 8 chord qualities: `major, minor, diminished, augmented, major7, minor7, dominant, half-diminished`. In the step four of the cleansing process above, we identify one chord per bar in a music piece. In the chord progression track of GETScore, we allocate the chord root in the first row and the quality in the second row. The chord track is entirely filled, without any paddings. Figure 4 is an example of GETScore with the chord track.

Figure 4: An example shows the GETScore with seven tracks used in our experiment, where the numbers denote the token indices. The example is for display only and does not correspond to a real-world music piece.

**Vocabulary**  In the last step of the cleansing process mentioned above, the construction of the vocabulary is essential before converting music fragments into GETScores. In GETMusic, each track has its own pitch vocabulary, while the duration vocabulary is shared among all tracks.

The maximum duration supported by the GETMusic is 16 time units, resulting in a total of 17 duration tokens ranging from 0 (the special duration token for drums) to 16 time units. To construct the pitch vocabulary, the music is first normalized to either the `C major` or `A minor` key, which significantly reduces the number of pitch token combinations. For each track, we identify the unique (compound) pitch tokens and rank them based on their frequency. During inference, the input music is also first normalized to `C major` or `A minor` and tokenized accordingly. GETMusic re-normalizes the generated music to its original key.

The final vocabulary consists of 17 duration tokens, 20 chord tokens, a padding token, a `[MASK]` token, an `[EMPTY]` token, and specific pitch tokens for each track: 128 for lead, 853 for bass, 4,369 for drums, 1,555 for piano, 3,568 for guitar, and 1,370 for strings. In total, the vocabulary consists of 11,883 tokens.

## C  HUMAN EVALUATION

We recruited a group of 10 evaluators who possessed a basic knowledge of music. They participated in a blind test where they were presented with songs generated by GETMusic as well as baseline models.

Before the blind test, we randomly selected five generated songs of each task and evaluated them ourselves. These scores were used as references for the evaluators and were not considered in the final results. What's more, some instructions were provided to ensure accurate evaluations. To evaluate the conditional generation, evaluators need to:

• Listen to the source tracks.

• Listen to the entire musical piece and evaluate its overall performance. How about the melodic and rhythmic qualities? How about the regularity in the music score? How about the coherence?

• Turn off the source tracks and assess generated target tracks independently. Do generated tracks remain melodic? Do they complement the source tracks effectively? Do you perceive a clear chord progression?

For the unconditional generation, the evaluators only needed to follow the second instruction, that is, listening to the entire musical piece. After completing the listening process, the evaluators rated the generated results on a scale from 1 (Poor) to 5 (Excellent), reflecting their overall judgment of the music.

To facilitate the evaluation process, the evaluators used Musescore (`https://musescore.org/en`). This software supported the necessary functionality of turning off and lowering specific tracks while providing music scores. Figure 5 shows the evaluation interface. All music files used for evaluation are anonymized. An evaluators was paid at an hourly wage of $8, and the entire rating process took approximately 10 hours to complete.

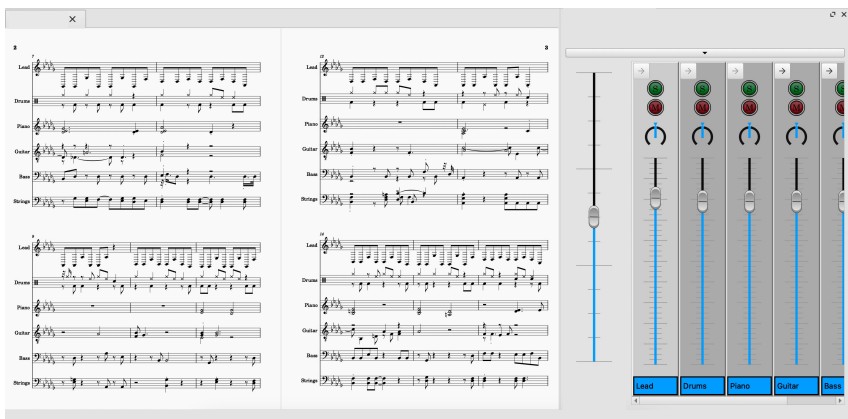

Figure 5: The Musescore interface.

