# OpenReview forum: "GETMusic: Generating Music Tracks with a Unified Representation and Diffusion Framework"
_ICLR.cc/2024/Conference — Submitted to ICLR 2024_

### Official Review · Reviewer_xCek · 2023-10-22

**Soundness:** 2 fair
**Presentation:** 1 poor
**Contribution:** 2 fair
**Rating:** 3
**Confidence:** 5

**Summary:**

This paper introduces GETMusic for multi-track symbolic music generation. GETMusic consists of a novel data representation (named GETScore) and a (discrete) diffusion model (named GETDiff). The data representation is a more condensed and meaningful way to represent multi-track data structure. The the non-autoregressive nature of diffusion models and the defined mask token naturally provide the model with the flexibility to do a lot of generation tasks, including music continuation, infilling and track generation based on arbitrary given tracks. Experimental results show the model is more versatile and generation quality outperforms the the baselines.

**Strengths:**

1. Originality: only a few studies use diffusion model in symbolic music generation. This is probably the first to use discrete version diffusion model in symbolic music. The idea is worth experimenting because it fits the goal of controllability in symbolic music generation.
2. Quality: From the demo page, the model learns the usual function for each instrument and shows coherency (though quite limited) in terms of inter-track relations and phrase structure.

**Weaknesses:**

In this paper, I believe there are some important concepts not explained in detail, which are crucial to understand the main contribution of the paper:

1. Discrete diffusion models. How is the model different from continuous diffusion models? Why is it more suitable for symbolic music generation? What is the relations between the proposed work and previous masked language models (e.g., Coconet[1], DeepBach [2] and MuseBERT [3]). More discussion can help better understand the approach and the expected generation quality. In fact, continuous diffusion in symbolic music generation (for single track) is quite successful [4, 5] and the current discrete version does not yield high generation quality (according to demo page) though the task is more difficult. It will be more clear if some explanation or comparison is made.
2. Condition flag. This is an interesting idea proposed in this paper and not in the original discrete diffusion model paper. Why is it necessary? Does it make musical sense? How is it operated and learned in detail? What strategy does it learn to “recover” the music? Without sufficient discussion, it is difficult to understand the ablation study.

Besides, the paper writing is not clear. Much content (mainly the general idea and big picture) is reinforced or duplicated too much throughout the paper, making the writing not concise and the details not clear. For example, section 1 and 2 both discusses sequence-based and image-based representation in almost same detailedness, and the “general” advantages are discussed in both section 1 and 3). If the paper is more concise, there will be more space for the concepts that should be explained in more detail. Also, the figure reference is not clear (figure 1 is not reference in the beginning. Figure 1 circle 2 and 3 seems similar). Table 1 and table 2 duplicate a lot.

Finally, the experiment result is not so convincing. I must admit music generation is very difficult to evaluate and the authors have already tried hard. Still, I have the following concerns which I think can be achieved in this study.

1. What aspects of generation quality is considered in this paper? If the goal is track generation based on others, one should analyze the interplay between tracks and show how the generated track follows the other tracks. (For example, what will the keyboard change if there is or is not a drum part?) Are the generated pitch coherent vertically? Are pitch ranges overlapping? There can be actually music sample analysis or proposed metrics for this. On the other hand, if the emphasis is for long-term generation, one should proposed metric or conduct analysis on phrase structure. Currently, all aspects are considered together making evaluation intractable.
2. Given the small number of participants, is the subjective evaluation result significant? What music aspects are they rating? Will the result be so noisy because all music from different baseline methods share nothing in common except length?
3. The music quality presented in the demo page is below expectation. It could be due to the difficulty of the multi-track generation task. If the generation result is way better than previous methods (in what sense), the work will be a significant improvement (in that sense). Otherwise the contribution is marginal. Some music analysis will help reader understand the paper better.

I hope the writing can be more careful. For example, in figure 1, the upper-left track name “melody” is not in the same font; the ellipsis in lower-left image is not placed correctly; the rectangles in the the lower-left image is not perfectly aligned; difference between circle 2 and 3 is not clear; the color scheme in the right column “Symbolic Music” has ambiguous meaning. “Transformer” should better be capitalized.

[1] Cheng-Zhi Anna Huang, Tim Cooijmans, Adam Roberts, Aaron C. Courville, Douglas Eck: Counterpoint by Convolution. ISMIR 2017: 211-218

[2] Gaëtan Hadjeres, François Pachet, Frank Nielsen: DeepBach: a Steerable Model for Bach Chorales Generation. ICML 2017: 1362-1371

[3] Ziyu Wang, Gus Xia: MuseBERT: Pre-training Music Representation for Music Understanding and Controllable Generation. ISMIR 2021: 722-729

[4] Gautam Mittal, Jesse H. Engel, Curtis Hawthorne, Ian Simon: Symbolic Music Generation with Diffusion Models. ISMIR 2021: 468-475

[5] Lejun Min, Junyan Jiang, Gus Xia, Jingwei Zhao: Polyffusion: A Diffusion Model for Polyphonic Score Generation with Internal and External Controls. CoRR abs/2307.10304 (2023)

**Questions:**

Q1: Why does the paper use the discrete version diffusion model? How do condition flags work and what strategy do the flags learn?

Q2: How is chord accuracy computed? During inference why are there ground-truth chords? Is it the case that only the last track is chord (so why use the row id i here in eq. 6)?

Q3: Evaluation. In what musical aspects do the proposed method outperform? Could the authors show some analysis examples or related metrics? Alternatively, could the authors show more convincing demos regarding the specific musical aspects.



Q4: GETScore is not a complete language in the sense that a small amount of important duration information is lost. How to solve the potential problem? Could it be a unified solution suitable for music data structure if the representation is lossy?

Q5: Interested to see what does the diffusion model do in the 100 diffusion steps. If the results correspond to meaningful music concepts (e.g., music hierarchy), it will be ideal to show them in the paper.

---

### Official Review · Reviewer_gxWR · 2023-10-31

**Soundness:** 3 good
**Presentation:** 3 good
**Contribution:** 3 good
**Rating:** 5
**Confidence:** 3

**Summary:**

- The paper presents GETMusic: a discrete diffusion-based approach (GETDiff) for symbolic music generation using a novel musical score representation for multi-track scores (GETScore).
- GETScore is a 2D token representation for symbolic music with two main features:
    - Tracks are arranged vertically, similar to musical scores, where each track is represented as 2 rows: pitches and durations. The horizontal axis is subdivided into a time-unit representing sixteenth notes.
    - Pitches are represented as a single token regardless of the number of pitches being played in a particular time-step. For example, a chord will be represented as a single token. These are called compound tokens. Drum instruments are also given specific pitch token IDs and placed on the score representation based on the location of the instrument hit, along with a special duration token that represent drum hits.
- The GETDiff process involves first adding mask tokens randomly selected tracks conditioned on other tracks based on a transition matrix in the forward process. Empty tokens are introduced to allow chosen tracks to be completely ignored. During the denoising process, the GETScore for the previous diffusion step is obtained through a Gumbel-softmax operation applied to the outputs of a roformer-based model.
- The GETMusic model is evaluated on three tasks: melody generation based on accompaniment, accompaniment generation based on melody, and generation from scratch. The authors train on score obtained from MuseScore, and compare against a couple of different baselines like the PopMAG and Museformer. Based on some objective evaluation metrics, as well as surveys from participants with some music knowledge the GETMusic model outperforms the other two models are all the three tasks.

**Strengths:**

- The results on accompaniment generation based on input lead melody are pretty convincing. The other results are also decent, but the aforementioned are higher quality.
- Authors demonstrated the utility of discrete diffusion for the task of symbolic music generation. In the past, diffusion models have been applied to music generation but typically it involves first learning a latent space of symbolic music and subsequently training a LDM model on top of the latent space (Mittal et al. 2021).

**Weaknesses:**

- The GETScore notation seems to have limited utility. While it works for score obtained from sources like MuseScore which do not typically contain expressive performances, encoding actual musical performance or even compositions with some expressive parameters seems tricky.
- There is existing literature on discrete diffusion models applied to symbolic music generation [1]. The authors are encouraged to comment on the differences between their work and the one mentioned here.
- The evaluation section is not very clear on what data is used for the objective and human evaluations. It is mentioned at the end of the Tasks and Baselines subsection that 100 samples of 128 beats are generated. On first reading it seemed like that is the setup used for task 3: generation from scratch. The authors are encouraged to clarify the specifics of evaluation for each of the 3 tasks. Additionally, it would be nice to have error bars for the human evaluations. The listener pool seems small and it’s not obvious if the differences between the baselines and the presented method is significant.
- The human evaluation protocol needs some work. See questions section for specifics.

[1] Plasser, Matthias, Silvan Peter, and Gerhard Widmer. "Discrete Diffusion Probabilistic Models for Symbolic Music Generation." arXiv preprint arXiv:2305.09489 (2023).

**Questions:**

- Are the human listening tests conducted with all the samples used for the objective evaluation for the 3 tasks as well? It is not mentioned in the paper exactly how many scores each listener examined.
- Why the listeners have to give 1 rating for so many questions? It looks like there are questions for overall quality, rhythmic quality, melodic quality, regularity, coherence for task 3 but eventually the listener enters a single rating. For task 1 and 2 as well, there are multiple questions but 1 rating.

---

### Official Review · Reviewer_NjSU · 2023-10-31

**Soundness:** 2 fair
**Presentation:** 3 good
**Contribution:** 1 poor
**Rating:** 3
**Confidence:** 4

**Summary:**

This paper presents a novel symbolic music representation (GETScore) together with a tailored diffusion model (GETDiff) for conditional music track generation based on that representation. In GETScore, individual polyphonic tracks are encoded as compact aligned sequences of pitch grouping and duration tokens. Later on, a discrete diffusion model is supervised with a mask-reconstruction objective. The well-trained model can flexibly generate rest musical tracks given existing ones and perform span-infilling. Objective and subjective evaluations are conducted against two previous models for conditional music track generation tasks. An ablation study is also conducted to validate the choice of the GetScore-GetDiff combo and other engineering techniques.

**Strengths:**

* GETMusic is a well-engineered generic framework for a range of conditional music track generation tasks. The paper is well-formated and generally clear to follow for the most part.

* The evaluation part features existing strong baselines and the online demo also sounds good.

* The code will be publicly available and can promote reproducibility and foster related research.

**Weaknesses:**

* Although the proposed model is well-engineered, it seems to lack novel insights from a general music representation learning point of view. Model-wise, the discrete diffusion model seems to be adopted without much insightful framework design. Representation-wise,  a conceptually similar BPE method for encoding note groupings has been previously considered [1]. Hence I am not convinced that there is a strong contribution here that would be of interest to the broader community of representation learning on music.

* Certain rationales of the representation design are not so intuitive to me. The emphasis on "665 source-target combinations" looks a bit awkward. Indeed all combinations are theoretically possible, but not all of them are commonly adopted in composition practice. Moreover, in real composition, a single type of instrument could be reused in a few tracks for different purposes, but the paper seems to merge all of them, which might make it less flexible or intuitive for certain users

* The evaluation results look strongly in favour of the proposed model and it would be better if the confidence interval could also be reported. This will help one convincingly tell that the results are statistically significant rather than out of luck.

[1] J. Liu, *et al*. Symphony Generation with Permutation Invariant Language Model. *ISMIR* 2022.

**Questions:**

* For computing chord accuracy *CA*, I understand that groundtruth chord $C_{i, j}$ is given as target. But how to obtain chord label $C_{i,j}^\prime$ from a generated track?
* Page 9 Line 1 mentions that the ablation experiment is conducted on the *generating tracks from scratch* task and *CA* is reported in Table 2. But *CA* is actually not applicable to this task. Should the task here be *accompaniment generation based on the melody*?

---

### Official Review · Reviewer_LUPw · 2023-11-01

**Soundness:** 2 fair
**Presentation:** 3 good
**Contribution:** 2 fair
**Rating:** 6
**Confidence:** 3

**Summary:**

The paper introduces GETScore, a novel data representation for modeling multi-track symbolic music. Building on this, the authors then construct a diffusion model tailored for generating symbolic music using the proposed data representation. The research demonstrates that utilizing this novel representation, in conjunction with the diffusion model, facilitates versatile generation capabilities, including the unique "any-to-any" part generation.

**Strengths:**

1. A novel data representation of symbolic music that supports separate and temporally aligned tracks.
2. The application of the diffusion model in the proposed data representation. Specifically, it gives the ability to achieve "any-to-any" generation throughout the infilling.

**Weaknesses:**

On a personal note, I feel the overall contribution of the paper could be further emphasized. While the new data representation is evident, the enhanced controllability seems intrinsic to diffusion modeling. It would be beneficial if the authors could more distinctly spotlight other innovative aspects or contributions within the paper.

**Questions:**

Considering the presence of both an embedding and a decoding module, I'm curious as to why the authors opted for diffusion in the data space as opposed to the latent space, a choice more akin to text-to-image models.

---

### Meta-Review · Area_Chair_F4kz · 2023-12-10

**Metareview:**

This paper presents symbolic music representation (GETScore) and generation with a diffusion model (GETDiff). The score is a 2D token structure with one axis for different instruments and one axis for time. The diffusion denoises masked tokens. The approach is interesting and the results are convincing (e.g. reviewers NjSU, xCek, gxWR), but fails to convince of the superiority of its specific design choices (all reviewers) and lacks novel insights (e.g. NjSU). In its current state, the paper would fit better in a specialised venue about music (ISMIR, or a workshop).

**Justification For Why Not Higher Score:**

Scope and novelty is limited. It's token to token 2D diffusion on a dataset of symbolic music.

**Justification For Why Not Lower Score:**

N/A

---

### Decision · Program_Chairs · 2024-01-16

Reject